# Impact of Air Taxis on Air Traffic in the Vicinity of Airports

**Nils Ahrenhold *** , **Oliver Pohling** and **Sebastian Schier-Morgenthal**

Institute of Flight Guidance, German Aerospace Center (DLR), Lilienthalplatz 7, 38108 Braunschweig, Germany; oliver.pohling@dlr.de (O.P.); sebastian.schier@dlr.de (S.S.-M.)

* Correspondence: nils.ahrenhold@dlr.de; Tel.: +49-531-295-1184

**Abstract:** The extension of the current urban transportation system utilising the third dimension by air taxi (AT) operations represents a potential solution for the congestion of metropolitan areas. A major asset for AT operations is the connection to existing airports enabling the access to multiple other transportation systems. This paper develops an analytical model for AT operations and their capacity impact on airports, exemplary for Hamburg airport. The model is developed, based on the results of a fast time simulation (FTS) considering multiple aspects, such as vehicle configuration and touchdown and lift-off areas (TLOF). Collectively, three integration methods were analysed, each of them impacting the conventional air traffic differently. The results show that an integration using the runway-system is not possible with five ATs per hour. Further methods allow an integration of up to 20 air taxis per hour. Additionally, an energy consumption analysis of the ATs is conducted. Finally, proposals are given for integrating ATs at an airport and further strategies to extend the analytical model. Through this work, a model to calculate and predict an AT's influence on the airside capacity of an airport is designed. This is an important step for the practical implementation of AT operations at airports.

**Keywords:** air taxi; airports; air traffic management; UAM; capacity impact; fast-time simulation; air traffic transportation

## 1. Introduction

Urban transportation systems are facing the challenge of congestion [1]. One reason is the steady growth of the population in cities. According to [2], two thirds of the world population will live in urban areas until 2050. This increased population density leads to a higher traffic volume and to congestion on the streets, especially during rush hours. The negative effects are not only higher noise levels and harmful emissions but also an increase of the so called door-to-door travel time. According to Flight Path 2050 [3], the current European mean travel time for 90% of the business and family travels, including one air segment, was five and a half hours in the year 2019. The goal is to reduce this time to four h.

One possible solution is the extension of the urban transportation system with an Urban Air Mobility (UAM) concept, using so called Vertical Takeoff and Landing (VTOL) vehicles as air taxis (AT) for transporting passengers within the city. In this way, a flexible point-to-point transportation possibility can be implemented. This possibility is not firmly tied to a two dimensional road system and may provide a solution for the capacity bottlenecks of the urban transportation. Therefore, UAM could contribute to the reduction of the targeted door-to-door travel time. To make a contribution and gain benefits, the main challenge is to connect the ATs to the existing transportation network, especially to the middle and long distance air transportation systems via airports.

This section will provide an overview of air taxi related literature. Divided into four main categories: vehicle design; touchdown and lift-off areas (TLOF); demand and scheduling; and operational constraints. The link of ATs to the existing air transportation system poses the question regarding VTOL design, integration methods within the airspace system

and airports as well as their impact on both. The challenges addressed at the beginning have already been dealt with intensively in research. A detailed analysis of the literature on UAM is provided by Biehle, Kellermann and Fischer in two published articles [4,5]. Possible use cases, as well as difficulties encountered during the implementation of urban delivery services and passenger transport are discussed. Holden and Goel [6] provide an analysis on VTOL market feasibility considering aspects of vehicle design, infrastructure operations and economics.

Shamiyeh and Rothfeld [7] investigated different personal air vehicle concepts for UAM, taking into account the new possibilities for distributed electric propulsion, opening ways for vehicle design. Therefore the characteristics, sizing effects and trade-offs between VTOL and cruise efficiency are studied. A comparison of the three main VTOL configurations (multicopter, lift+cruise and tilt wing) was conducted by Kraenzler and Schmitt [8], using a tool based on aerodynamic modules to provide a basis for decisions for vehicle design. Results show that the multicopter configuration especially can serve with an excellent hover performance in comparison with the other configurations. Therefore, the multicopter is particularly suitable for short-haul missions. Selecting the right configuration can have large effects on the mission structure and energy consumption.

Besides the central vehicle design, the TLOF development plays a role in implementing AT operations. Vascik and Hansman [9] investigated the vertiport capacity envelopes, different vertiport designs and operational parameters, depending on AT throughput. Within the study, a distinction was made between linear, satellite, pier and remote apron topology. They found that the ratio of gates to the TLOF at the vertiport is the design key factor. Multiple TLOFs on an vertiport might support simultaneous operations. Additionally, the Cologne Bonn airport, in cooperation with the RWTH Aachen university, presented a feasibility study for AT operations at Cologne Bonn airport [1]. This study analysed the requirements and challenges for an AT service. The focus was on the infrastructure, especially on potential locations for TLOF, the airspace structure and passenger handling. Passenger handling was only conducted for departure operations, while assuming no airside delay by a smoothly running integration of the ATs into controlled airspace due to a predefined separation minimum. Hence, the focus was on ground infrastructure. The main outcome showed that specific regulations for AT operations are still missing. A general integration of ATs at Cologne Bonn airport seems possible, although the simulation results showed that the potential bottleneck for integrating ATs into airports, which can introduce delay, may be on the airside and not on the landside of the airport.

The third category is the estimation and scheduling of the demand. This is needed to integrate the AT on-demand service into the market. Rothfeld and Antoniou [10] carried out a survey for the area of Munich (Germany) about the respondents' behaviour in terms of the choice of transportation, subdividing the transportation into public transportation, private car, autonomous taxi and autonomous AT. The survey points out that the parameters of travel time, travel cost and safety are the central concerns for acceptance. Rajendran and Zack [11] conducted a study that estimated the AT demand based on the current conventional taxi demand. Results show that AT on-demand service is appropriate when the travel time of AT is a minimum of 40% shorter than conventional travel time, depending on the AT maximum velocity (148 kt) and range (193 km).

The fourth category includes the operational constraints for ATs' adoption in current airspace management. In 2006, the National Aeronautics and Space Administration (NASA) published a study investigating the effects of AT operations on commercial air traffic and the impact on the predicated general delay of conventional air traffic for 2025 [12]. The study considered only fixed-wing, jet powered aircraft operating at non-towered or non-radar airports. Yet, no specific integration methods for ATs were proposed, only fixed-wing standard procedures were applied. The study was mainly focused on general vehicles, not VTOLs. The results showed that jet AT operations could possibly increase the en-route delay of the conventional air traffic and claim for new airspace structures. Vascik and Hansman identified a list of operational constraints for UAM operations [13], including

factors such as noise, public acceptance, accessibility of TLOF and interactions with air traffic controllers (ATC). Further, they applied the developed list to three U.S. cities [14]. The results showed that the critical constraints are noise acceptance, availability of TLOF and scalability of ATC. Potential integration methods and possible scaling constraints for UAM operations were further analysed [15]. Different integration locations in comparison to the runway system and operational concepts were discussed. The results indicated that segregated airspace cutouts for UAM flights are a possible concept for UAM operations at airports.

To summarize, there is a wide range of AT related literature for vehicle design, vertiports, demand and operational concepts. However, the present literature review identifies a lack of investigations into the impact of ATs, especially on the airside capacity of airports. Therefore, a concept or system is needed as a driver for implementation, to measure and estimate the impact for airports.

The challenge of integrating ATs into airports is the close and probably simultaneous operation of ATs and conventional air traffic. As these close operations might induce restrictions and separation issues, airport capacity can be affected. Depending on the AT vehicles, the traffic density and the integration of the ATs in the airport environment, different restrictions and separation rules have to be applied. A general understanding of these factors and their impact on the airside capacity is necessary for a successful and effective integration of UAM into airports.

This paper addresses two main research questions regarding UAM operations:

1.  How does the integration of ATs impact the airside capacity of a specific airport?
2.  How can this impact be generalised for airports willing to integrate ATs into their air transportation system?

Therefore, this study developed a fast time simulation (FTS) model for ATs operating within the jurisdictions of the Hamburg airport. The operation was evaluated by applying punctuality and energy criteria. Conflict points between ATs and conventional air traffic were identified, analysed and abstracted to general conflict points, eliminating specific circumstances which apply only for the use case of Hamburg airport. Finally, a concept was proposed including essential considerations for integrating ATs into air transportation systems and the vicinity of different airports. The present study designed a rating system for the impact of ATs on the airside capacity of airports.

## 2. Materials and Methods

The capacity model to evaluate the impact of AT integration on airside capacity was designed for Hamburg airport. Hamburg was chosen due to being a German traffic hub with some major roads, and having a traffic load of up to 200%. The model calculated the AT's movements, serving as an airport shuttle, as well as the conventional air traffic to and from Hamburg airport. Hamburg airport consists of two intersecting runways and a terminal on the south east side. The model flight plan for the conventional air traffic is built upon on the Demand Data Repository 2 (DDR2) data of the European Organisation for the Safety of Air Navigation (EUROCONTROL) for the year 2019 with the configuration: departures (RWY33) and landing (RWY23). In the further course, the model will be developed in separate steps. First, different model components, such as vehicle configuration and separations minima, will be defined. Afterwards, the simulation setup is discussed and evaluation criteria are introduced. Finally, the methods for analysing the results are explained.

### 2.1. Model Components

As a first step, the AT's vehicle configuration was chosen. It was chosen based on the selected use case. Since the ATs are functioning as an airport shuttle, they are on a short-haul mission. For that purpose, a multicopter configuration was chosen. The multicopter configuration consumes the least energy for a short-haul mission, due to the best hover performance compared to vectored-thrust and lift+cruise configuration [16]. The following

Table 1 provides an overview of the considered vehicle performance parameters. The parameters were defined based on a comparison of two public multicopter configurations (Volocopter-Germany and E-Hang-China [16]).

**Table 1.** Performance data of the modelled multicopter.

| Performance Data | |
| --- | --- |
| **Category** | **Value [Unit]** |
| Maximum Takeoff Weight | 450 [kg] |
| $V_{cruise}$ | 54 [kt] |
| Rate of Decent | 9.8 [ft/s] |
| Rate of Climb | 9.8 [ft/s] |
| Range | 50 [km] |
| Wake Vortex Category | Light |

Within the airport area of Hamburg, three different integration methods for VTOL TLOFs were identified: vertiport west (VPW), vertiport east (VPE) and runway (RWY), which are explained in the following. Those three possibilities will be called the integration method in this study, since they include an integration location as well as a linked integration procedure. Figure 1 displays a top view of Hamburg airport including the different integration methods and air taxi flight routes.

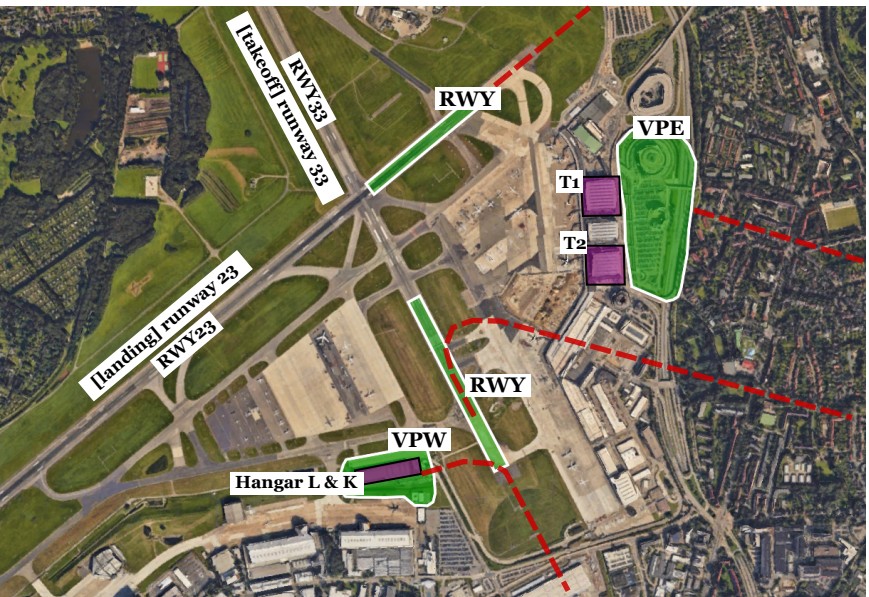

**Figure 1.** Top view Hamburg airport including integration methods for air taxis; RWY33 from south to north; RWY23 from northeast to southwest, takeoff and landing areas for air taxis, vertiport east (VPE), vertiport west (VPW) and runway (RWY): green with white borders; terminal 1, 2 (T1, T2) and hangar L, K: purple with black borders; air taxi flight routes: red broken line.

VPW: The first integration method uses a TLOF, so-called vertiport, at the airport. It is indicated with the label *VPW* in Figure 1. The *VPW* is located west of the RWY33 on the airside of the airport, using hangar L and K as a possible location. This vertiport is partially dependent on the conventional air traffic, caused by the parallel flight path.

RWY: The second integration method uses the conventional runway system and taxi ways of the airport, labelled as *RWY* in Figure 1. This integration method introduces full dependency on the conventional air traffic.

VPE: The third integration method considers a vertiport that is close to the conventional air traffic but a non-parallel flight path. This vertiport is located on the parking level

east of terminal one and two and labelled by *VPE* (see Figure 1), taking into account a minimum angle of 15 ° between the flight path of the conventional air traffic and the AT flight path. Hence, an independent operation is possible [17].

Next, after defining the TLOF at the airport, possible vertiport locations within the city are selected after a parameter study. The following parameters are used for selection based on the indicators for mobility factors in Germany [18]:

- population density;
- distribution of income;
- transportation link;
- area classification.

The AT service is predicted to be an extension of the urban transportation system through its flexible and fast connection purpose. Therefore, areas with a high population density and underdeveloped public transportation connection to the airport are sought. In addition, the costs for an AT service are estimated to reach higher price segments. The willingness to use such an AT service will be significantly dependent on the income. Thus, areas with a higher income are additionally selected. Altogether, the following eight areas within the Hamburg city area were selected: Blankenese, Finkenwerder, Harburg, HafenCity, Central Station, Bergedorf, Rahlstedt and Quickborn. Figure 2 displays their location within Hamburg city.

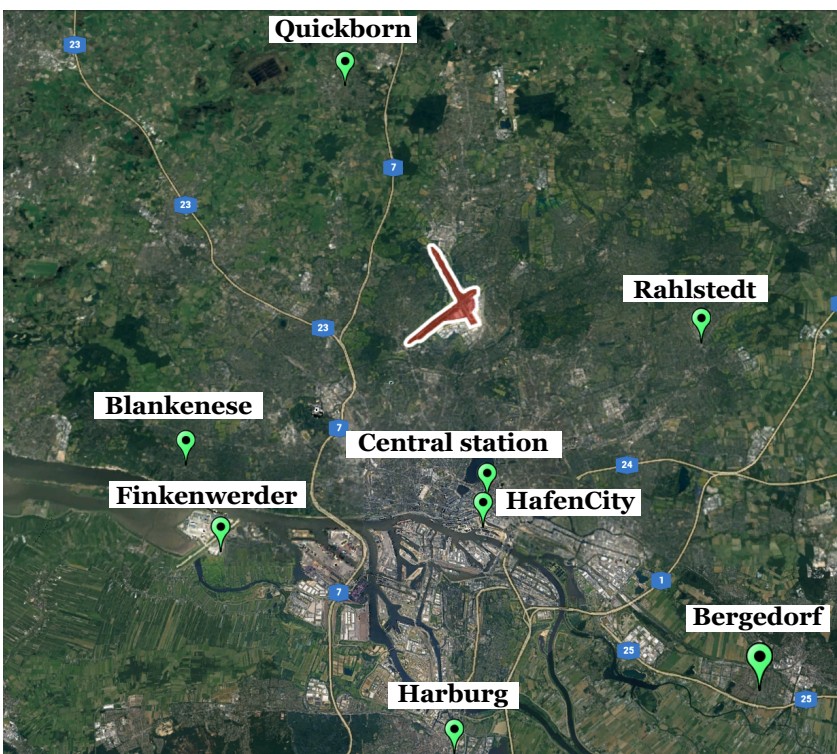

**Figure 2.** Top view of Hamburg city including takeoff and lift-off areas (TLOF) locations Blankenese, Finkenwerder, Harburg, HafenCity, Central Station, Bergedorf, Rahlstedt and Quickborn.

The flight rules and routes were designed for a safe and reliable integration of the ATs into the air transportation system. The concept of choice was the hemispheric corridor concept [19]. The ATs operate under visual flight rules (VFR) in a flight corridor within 1000–2000 ft. The lowest flight altitude represents the minimum safety altitude for VFR flights over cities [20]. Separations from 200 ft vertical and 2000 ft lateral between ATs are applied based on the assumption that two ATs flying towards each other, with the assumed velocity, need approximately ten seconds for collision avoidance manoeuvres [21]. Additionally, wake turbulence separations are used between ATs and conventional aircraft.

The AT flight routes from the city-TLOF to the airport-TLOF were designed for all three integration methods separately. They are modelled based on the VFR compulsory reporting points for Hamburg controlled air space and the Specific Operations Risk Assessment (SORA) concept of the Joint Authorities for Rulemaking on Unmanned Systems (JARUS) [22]. The SORA concept represents a risk analysis for the usage of unmanned flight systems. Regarding the current status, the SORA concept is only applicable for drones of a 'specific' category. It was estimated that ATs will be in the 'certified' category Operation Type 2 (unmanned) or Operation Type 3 (manned), the general approach for designing the AT flight route in the vicinity of conventional air traffic consisted of three basic step: first a lateral separation was applied, second a vertical separation was applied and if both were not suitable, holdings were introduced. This is the first approach to design and illustrate AT flight routes within the city. The study did not focus on the optimal routing problem. The route definition includes optimisation potential, which should be qualified by further studies. During the design process, five main conflict points (CPs) for the AT flight routes within the three integration methods (*RWY, VPW, VPE*) occurred. Those CPs are displayed in Figure 3 and will be discussed in the following.

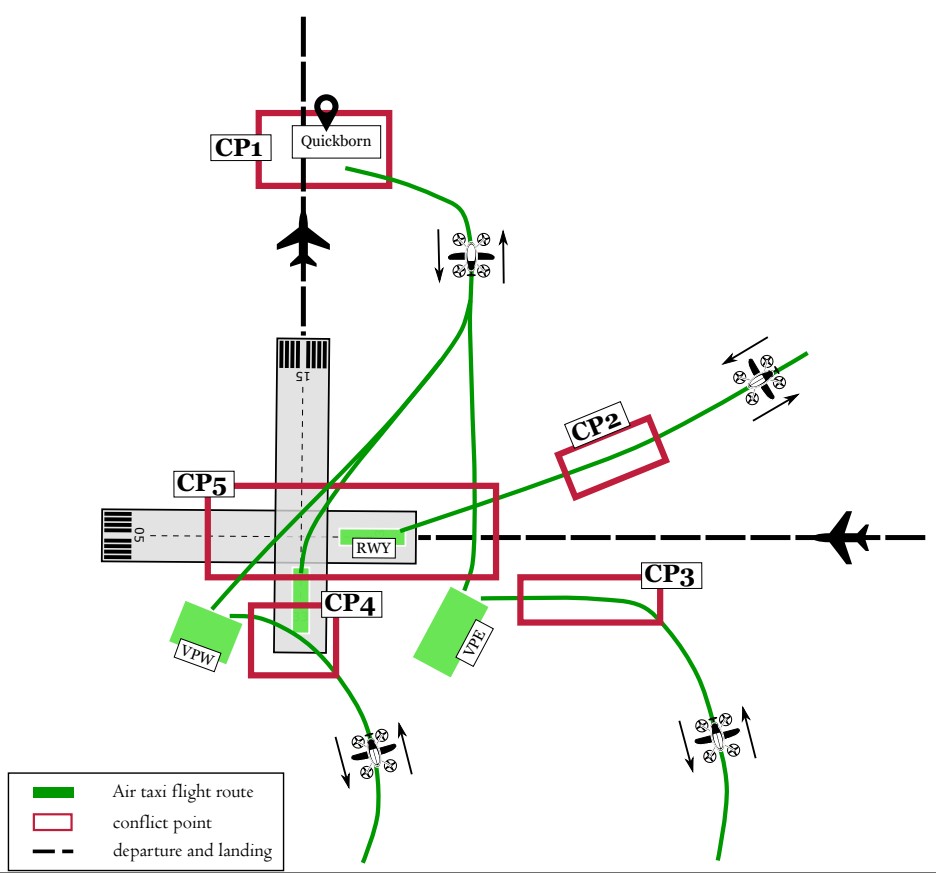

**Figure 3.** Conflict points (CPs) for air taxi integration at Hamburg airport via different takeoff and lift-off areas; air taxi flight routes: green line; conflict points: red box; conventional departure and landing routes: black broken line.

1.  Quickborn—CP1 (RWY, VPW, VPE): The northern TLOF-vertiport (Quickborn) is situated in the departure area of the RWY33. After analysing aircraft movement data from the EUROCONTROLS' DDR2 repository, it was found that it was not feasible to fly safely around this area since the departure routes disperse into eastern and western routes. An altitude restriction serves as a solution to avoid conflicts in vertical separation. The ATs were restricted to fly at their minimum altitude while conventional departing traffic already had an altitude greater than 2000 ft to secure minimum separation.

2.    Final Approach—CP2/CP3 (RWY): The integration of ATs into the final approach of the conventional air traffic asked for sufficient separation to cope with different approach speeds (ATs approx. 100 kt slower than conventional aircraft). The objective was to achieve a low impact on the separation of the conventional air traffic. Therefore, an integration point was chosen close to the touch-down (TD) of the runway. The capability of the ATs to fly a small curved radius enabled an integration point approximated 1 NM before the TD.

3.    Jet Blast—CP4 (VPW): For VPW, dependencies between the ATs and conventional air traffic have to be taken into account. Regarding ICAO, the TLOF-vertiport (treated as a helipad) and runway need to have a minimum distance of 200 ft according to the AT's maximum takeoff weight [23]. The minimum defined safety distance was violated within the model. Thus, an aircraft's jet blast effects had to be taken into consideration. Therefore, a model for the jet blast was applied to predict the necessary minimum separation for operations on VPW. Since the aircraft of the heavy category were also operating at Hamburg airport, the jet blast length (l) and width (w) were estimated to be $l = 1000$ ft and $w = 100$ ft for a B777. The jet blast of a small or medium aircraft was estimated to be smaller and less harmful. According to Slaboch [24], the distribution time after that no jet blasts of a B777 have affected a subsequent aircraft, is estimated to be 62 s with a 95% confidence interval. This already includes a factor of safety. Transferred to the AT approaching velocity, a minimum distance of 0.5 NM was defined for a safe starting and landing process.

4.    Northern Routes—CP5 (RWY, VPW, VPE): The AT flight routes northwards need special attention. Since Hamburg airport consists of a configuration of two intersecting runways, all ATs flying northwards or coming from the north have to cross the final approach of RWY23. The flight performance of the ATs prevents a feasible flight over, with a vertical safety separation large enough to not interfere with the conventional air traffic. Safety separations could be violated, which may causes go-arounds of the fixed-wing aircraft. Further, a crossing without sufficient vertical separation evokes delay during the final approach and uneconomical spacious separations. One possible solution is an extension of the departure flight routes towards the northeast with a fly under in the vicinity of the final approach fix (FAF). Here, a sufficient vertical separation of 1000 ft can be enabled. The flight distance increases by approx. 30% in comparison to the direct path with a fly over. It is still within the range of the defined ATs and still serves a faster connection (approx. 25 min) compared to the public transportation (approx. one hour). ATs coming from the north are not extended to the east, but are integrated into the final approach of the conventional air traffic. After a short flight distance, the ATs leave the final approach and head back to the AT flight route regarding the integration method.

After defining airport/city-TLOF, flight rules and routes, the design process of the traffic scenario will be explained in the following. The applied traffic scenarios were designed for validation purposes, to show the interaction of the different airspace users and to evaluate the impact of the ATs on the airside capacity. Therefore, a reference, the so called baseline scenario (BS), was defined. The BS includes a complete day of conventional air traffic service (24 h) exclusively at Hamburg airport. Based on aircraft movement data from EUROCONTROL's DDR2 repository, the conventional flight plan was modelled for the defined runway configuration: departure (RWY33) and landing (RWY23). This operation configuration was the most used configuration during the year 2019 at Hamburg airport with 46% overall usage time [25]. Figure 4 displays the defined flight plan of the conventional air traffic for 24 h.

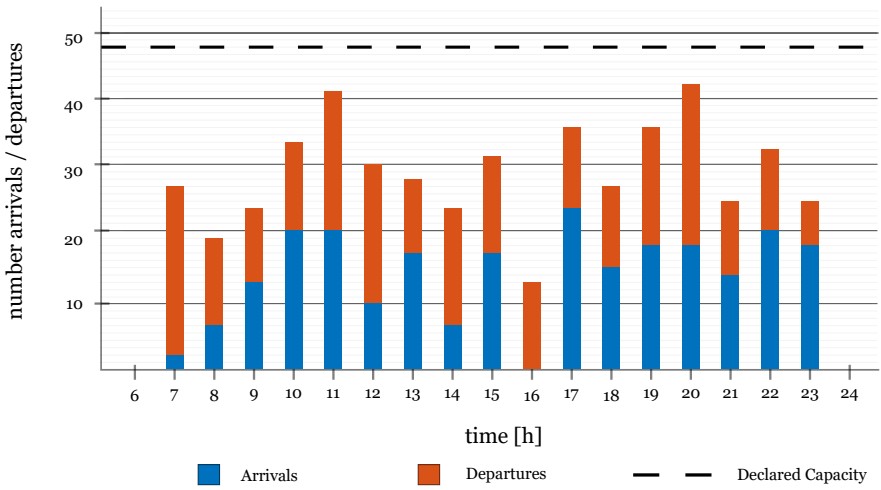

**Figure 4.** Flight plan for conventional air traffic used in simulation based on Demand Data repository 2 (DDR2) from 19.09.2019; total of 239 arrivals and 247 departures; declared capacity for Hamburg airport 2021 from [26].

Two experimental conditions were determined with independent variables for the analysis: The integration method (*VPW, RWY, VPE*) and the number of integrated ATs per hour. For every integration method, a defined number of ATs is inserted into the BS separately and increased continually, starting with a maximum number of five ATs per hour up (*AT5*) to twenty ATs per hour (*AT20*). The maximum number is chosen according to Crossley and Mane [27]. The study estimates that an AT service with a fleet size of 21 ATs could operate profitably as a transportation service within a city. Under the premise of comparable city sizes, the estimations are applicable. Figure 5 illustrates the AT flight plan for increasing demand from five ATs per hour (AT5) up to twenty ATs per hour (AT20). Since the AT service is expected to be largely on-demand driven, the ATs flight plan is directly linked to the conventional flight plan. The AT on-demand service is needed after passengers get out of the plane and receive their luggage. Accordingly, the AT distribution was conducted compared to the conventional flight plan and with a time shift of 30 min to conventional departure and arrivals.

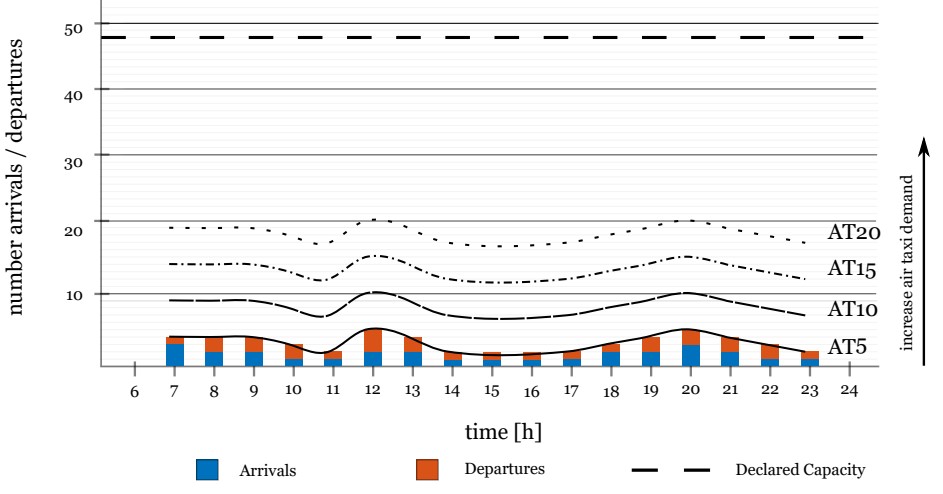

**Figure 5.** Flight plan for air taxi used in simulation with incremental increase of demand from AT5 (five air taxis per hour) up to AT20 (twenty air taxis per hour); declared capacity for 2021 from [26].

### 2.2. Simulation Model

In this paper, the traffic simulation was performed with the FTS tool AirTOP, which was developed by Airtopsoft. AirTOP enables a gate-to-gate traffic simulation in consideration of various rules that are executed by so-called agents [28].

To set up the simulation model, AirTOP needs different input data. Two central input variables were the airport/airspace layout and the traffic data. The simulation model includes the ground layout of Hamburg airport and its related terminal manoeuvring area (TMA) according to the German Aeronautical Information Publication (AIP). The airport model includes runways, taxiways and aircraft stands, whereas the TMA consists of standard instrument departure routes, standard terminal arrival routes, transitions and the approach path. Regarding the traffic data, a flight plan is the essential part of the traffic simulation. AirTOP's minimum requirements for a flight plan are a callsign, an aircraft, a time value (injection time) and a routing. As mentioned in Section 2, traffic data of the conventional aircraft were obtained through EUROCONTROL's DDR2 repository. The ATs' traffic data were developed and adjusted to the conventional flight plan (cf. Section 2.1). In terms of the flight performance, the simulation model utilises performance characteristics of EUROCONTROL's base of aircraft data (BADA) (version 3) [29] for the conventional aircraft to simulate the airside motion. The AT performance model uses the characteristics displayed in Table 1.

Another relevant aspect is the simulation framework. All simulation runs were executed under Instrument Meteorological Conditions (IMC). Wind was neglected. The study provides a first estimation under best weather conditions. Valid data for crosswind behaviour of the air taxis are not yet present. Furthermore, the conventional arrival aircraft enter the simulation at their last en-route waypoint that leads to the initial approach fix, whereas the departing aircraft leave the simulation after reaching their first en-route waypoint. However, the AT flights were implemented completely from their takeoff location to their destination due to their short flight distance compared to the flight distance of conventional air traffic. Additionally, a turnaround (aircraft or AT rotation) was not conducted. Hence, each simulated aircraft or AT only conducted one flight.

Concluding this section, the simulation model considers the dependencies between the ATs and the aircraft specified in Section 2.1. In order to achieve this, the TLOF-vertiports were modelled as a helipad with dimensions following already existing helipads at Hamburg airport. The FTS is able to respect dependencies between the runway system and the TLOF-vertiports. As mentioned in Section 2.1, the AT operations within the VPE integration can be assumed as independent from the conventional air traffic. For the VPW integration, a dependency occurs with respect to departures from RWY33. Here, two conflicting combinations can occur: (1) departing aircraft with departing AT; or (2) departing aircraft with arriving AT. The first situation is handled by a first-come-first-serve (FCFS) strategy. Thus, all flights are handled equally. The first aircraft or AT that requests takeoff clearance will be served first. In terms of (2), a different handling is applied. A departing aircraft from RWY33 will get takeoff clearance only if an arriving AT is at the required distance (cf. Section 2.1) away from the threshold RWY33. Moving on to the RWY integration, there are two conflicting combinations: (1) between departing aircraft and AT; and (2) between arriving aircraft and AT. In both situations, the traffic flows are handled equally and the FCFS strategy is used. In contrast to the handling of departures on the same runway, the integration of aircraft and ATs to a common final approach is much more difficult because of the severe performance differences. The FCFS strategy is realised by holding patterns. Aircraft use the already existing holding patterns inside the TMA, whereas separate holdings are created for ATs. These holdings are modelled following the ones for aircraft. However, the AT holdings have a shorter leg timing of 15 s instead of 1 min as a result of the lower speed. An aircraft has to enter a holding if an AT has passed a specific point on its approach path and vice versa, which does not allow for a parallel operation anymore.

The arrival separation minima (cf. Section 2.1) are established among the ATs via speed adjustments and a holding pattern prior to the final approach. This is necessary due to a lack of horizontal sequencing possibilities. Regarding the conventional air traffic, the separation standards according to the ICAO Doc 4444 [30] and the German AIP were considered.

### 2.3. Evaluation Criteria

The integration of ATs into the airport will lower airport capacity due to the identified dependencies. This decrease is acceptable as long as the transportation targets are not violated. Safety, as the primary target, is assumed to be assured continuously due to the applied separation minima. Punctuality, as the second target, will be used as a criterion by which simulation results can be quantified. Comparing the number of ATs meeting the punctuality criteria under different AT traffic volumes, a feasible number of integrated ATs can be derived. Table 2 provides an overview of commonly used punctuality criteria for different transport carriers.

**Table 2.** Summary of different punctuality criteria [31–33].

| Definition of Punctuality | | |
|:---:|:---:|:---:|
| **Name** | **Definition** | **Evaluation** |
| Aviation | | |
| EUROCONTROL | $\geqslant 15\,\text{min}$ | delayed |
| Long-distance traffic | | |
| German railway | $\geqslant 5\,\text{min}\,59\,\text{s}$ | delayed |
| Public transit | | |
| Leipzig (train) | $\geqslant 3\,\text{min}$ | delayed |
| Hanover (train) | $\geqslant 3\,\text{min}$ | delayed |
| Hanover (suburban train) | $\geqslant 2\,\text{min}$ | delayed |
| Hanover (bus) | $\geqslant 5\,\text{min}$ | delayed |

Regarding the conventional air traffic, the 15-min criterion for the accepted total delay of an aircraft flight is selected. The 15-min criterion represents the Air Traffic Flow Management (ATFM) time slot from EUROCONTROL [33]. If this slot is violated, cumulative delay can occur, which takes a long time to dissolve within the air transportation system. A punctuality or accepted delay criterion for ATs is not present yet. Therefore, a new punctuality criterion is defined based on a comparison of different transport carriers. The greatest comparability exists with the public transit by reason of range and usage. Consequently, a 3-min AT punctuality criterion was defined, compared to the punctuality criteria for public transit (train) in Leipzig and Hanover (see Table 2). With a delay of three minutes, an AT is considered to be delayed. The numbers of punctual ATs and aircraft are calculated by comparing the BS takeoff and landing times with the takeoff and landing times from the experimental scenarios. Additional actual and estimated takeoff and landing times were compared with moving hour windows to evaluate where and when the impact of increased AT traffic causes a delay to the conventional air traffic side.

### 2.4. Energy Treatment in Fast Time Simulation Model

Besides the impact of ATs on the capacity, a short outline about the impact of ATs on the fuel consumption and $CO_2$ emissions of conventional air traffic is given in this paper. Based on the ICAO Doc 9883 [34], which lists the environmental impact as an important key performance area regarding the air traffic management performance, emissions are considered as an indicator of the environmental impact and need to be monitored. Additionally, a deeper look at the ATs' energy consumption was taken to examine if its battery capacity constitutes a bottleneck. The study focused on the fuel consumption and correspondent $CO_2$ emissions as one of the main environmental factors. The noise

emissions and additional factors will be part of a further study. In this section, the aim is to describe the setup of energy treatment within the FTS model.

Since the aircraft's performance characteristics were obtained from BADA (cf. Section 2.2), AirTOP was able to calculate the fuel consumption if the aircraft data contain values for the fuel flow. However, the ATs are operated electrically, which introduces a novel way of energy consumption requiring a new definition of "fuel flow". A fuel flow equivalent, $FF_{eq}$, is needed for the ATs so that the FTS tool is able to calculate the consumption during the operation. The $FF_{eq}$ can be calculated by transforming the energy consumption per flight time with the energy density. Thereby, the flight time was cancelled and the equation was simplified to the following formula:

$$FF_{eq} = \frac{P}{60 \, \text{min/h} \cdot e}. \tag{1}$$

Here, $FF_{eq}$ denotes the fuel flow equivalent, $P$ the power and $e$ the energy density of jet fuel. This conversion is possible by using the energy density of jet fuel [35], $e = 42.8 \, \text{MJ/kg} \approx 11.9 \, \text{kWh/kg}$. The energy consumption of ATs depends on the used power. Shamiyeh et al. [7] compiled an overview of relevant AT characteristics with three power specifications regarding the Volocopter 2x AT:

- $P_{Installed} = 75.6 \, \text{kW}$;
- $P_{Hover} = 39.5 \, \text{kW}$;
- $P_{Cruise} = 25.2 \, \text{kW}$.

However, these power values are not linked to a speed, which is why further assumptions are needed to describe different flight phases and performance characteristics at different altitudes. AirTOP requires altitude specific fuel flow values for the flight phases climb, cruise and descent to calculate the fuel consumption. Therefore, power values were assigned to selected altitudes, which can be seen in Table 3.

**Table 3.** Altitude specific power values and fuel flow equivalent (AT).

| Power Values for Air Taxi | | | |
|---|---|---|---|
| **Altitude** | **Norm Speed** | **Power** | **Fuel Flow Equivalent** |
| Climb | | | |
| 50 ft | 1 kt | 75.6 kW | 0.11 kg/min |
| 500 ft | 35 kt | 75.6 kW | 0.11 kg/min |
| 2000 ft | 50 kt | 39.5 kW | 0.05 kg/min |
| Cruise | | | |
| 500 ft | 50 kt | 25.2 kW | 0.04 kg/min |
| 2000 ft | 50 kt | 25.2 kW | 0.04 kg/min |
| Descent | | | |
| 0 ft | 20 kt | 75.6 kW | 0.11 kg/min |
| 500 ft | 25 kt | 75.6 kW | 0.11 kg/min |
| 1000 ft | 30 kt | 39.5 kW | 0.05 kg/min |
| 2000 ft | 50 kt | 25.2 kW | 0.04 kg/min |

This allocation was carried out under the premise that climb and descent procedures require more power, especially at lower altitudes. The results of Equation (1) are shown in Table 3. AirTOP is able to consider only two decimal places, which is why an appropriate rounding of the values is necessary. Because of the limited performance specifications, the results in Table 3 provide only a first estimation of the energy consumption.

## 3. Results

### 3.1. Punctuality Results

Figures 6 and 7 provide an overview of the number of delayed aircraft and ATs under the examined experimental conditions using, separately, the 15-min and 3-min punctuality criteria. When considering the integration of the conventional runway system (*RWY*), eight aircraft out of 486 in 24 h show a delay equal to or bigger than 15 min for an integration of a maximum of five ATs per hour. With increasing AT traffic, the number of delayed aircraft rises nearly linearly. Integrating a maximum of 20 ATs per hour delays 82 or 16.9% of all aircraft. A comparable behaviour can be seen in the ATs, regarding the 3-min criterion. Integrating five ATs per hour produced a total of twenty delayed ATs over 24 h. This number of delayed ATs increases to 48 or more than one third (35.5 %) of total included ATs in the *RWYAT20* scenario.

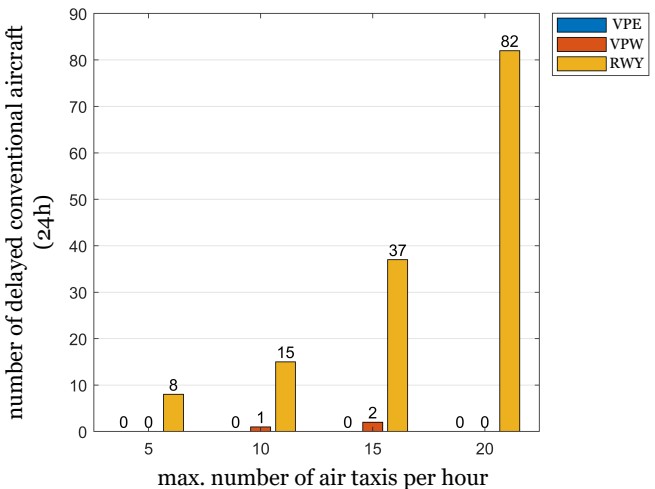

**Figure 6.** Number of delayed conventional aircraft during 24 h.

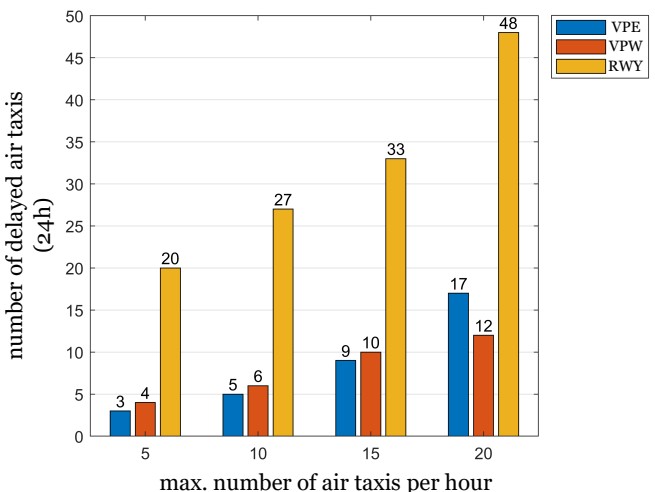

**Figure 7.** Number of delayed air taxis during 24 h.

When integrating ATs over *VPW*, the data indicate that a maximum of two aircraft are delayed for the *VPWAT10* and *VPWAT15* scenario. The number of delayed ATs is lower in comparison to *RWY* integration. It starts with four ATs (*VPWAT5*) and increases continuously, ending with twelve delayed ATs (*VPWAT20*).

When integrating ATs over *VPE*, the lowest impact can be reported. No aircraft displays a delay of 15 min or higher due to AT integration. The numbers of delayed ATs for *VPE* integration are lower than for *VPW*, except for the *VPEAT20* scenario.

### 3.2. Energy Results

The outcomes of the energy analysis will be explained in this section. First, a deeper look at the impacts of ATs on the $CO_2$ emissions and fuel consumption of aircraft is carried out. The relation between $CO_2$ emissions and the used fuel calculated by AirTOP is described as follows: under ideal circumstances, the combustion of 1 t kerosene releases 3.15 t $CO_2$ [36]. Figure 8 shows these impacts as a function of the integration method (*VPW*, *RWY*, *VPE*) and the maximum number of ATs per hour summarized for 24 h. Regarding the $CO_2$ emissions, Figure 8 reveals that the common usage of the runway system leads to an increase of the aircraft's $CO_2$ emissions compared to the BS scenario. Additionally, the $CO_2$ emissions increase with a higher number of ATs per hour, resulting in 34% higher total emissions for the case of 20 ATs per hour. In contrast, the integration methods *VPE* and *VPW* show, due to their dependencies that are lower than those of the *RWY* integration, only a very slight increase of about 1% in the $CO_2$ emissions. An increase as a function of the AT volume could be noticed, nevertheless it is very marginal (maximum increase of 0.2% because of the AT volume).

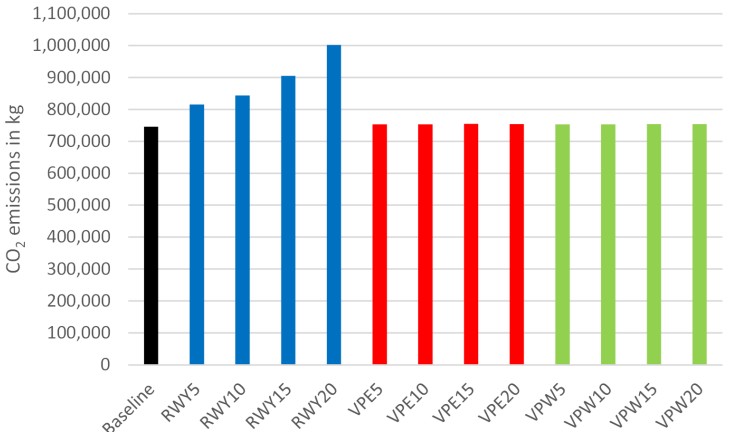

**Figure 8.** Total $CO_2$ emissions for conventional air traffic; blue = *RWY* integration, red = *VPE* integration, green = *VPW* integration.

Since $CO_2$ emissions are directly linked to the fuel consumption, both results ($CO_2$ emissions and fuel consumption) show the same shape and dependencies.

As mentioned earlier, an analysis of the ATs' energy consumption is important for examining the battery capacity. AirTOp calculated the equivalent used fuel of the ATs. These results were subsequently reconverted to energetic values. Figure 9 shows the total energy consumption for each AT integration method and traffic volume. It can be seen that the *RWY* integration leads to the highest energy consumption among the ATs. In terms of evaluating the battery capacity, Figure 10 presents the results regarding the energy reserve of the ATs. The first important indicator for evaluating the suitable integration method is if there are flights that exceeded the available battery capacity. It can be seen that this is only the case for the *RWY* integration. The number of affected ATs rises with a higher number of ATs per hour, although the scenario *RWYAT20* has fewer affected ATs than the scenario *RWYAT15*. The *VPW* integration causes up to 12 ATs with less than 10% of energy reserve.

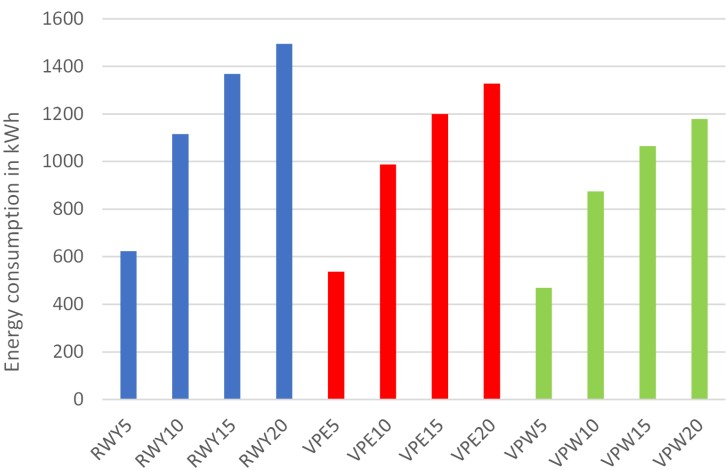

**Figure 9.** Total energy consumption for air taxis; blue = *RWY* integration, red = *VPE* integration, green = *VPW* integration.

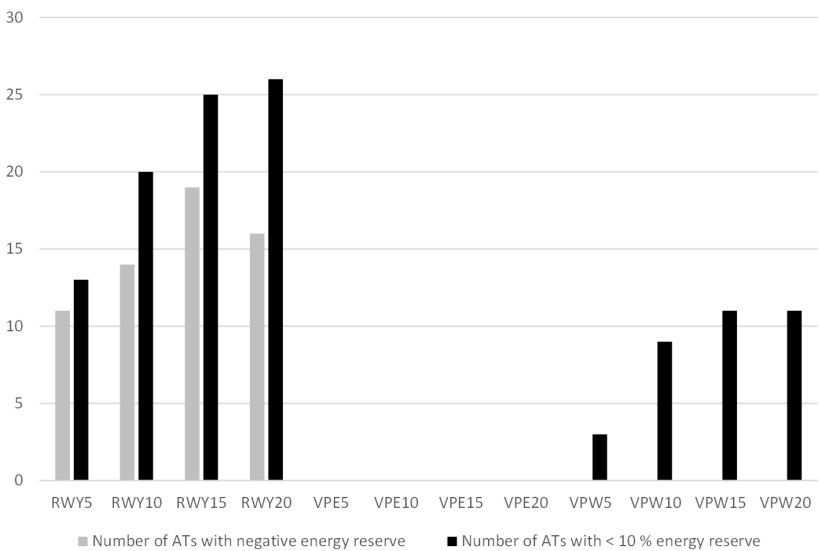

**Figure 10.** Energy reserve for air taxis.

## 4. Discussion

The present study designed and evaluated an FTS model to integrate ATs at Hamburg airport. The focus was on the impact on the airside capacity of the airport. Thereby, different concepts for AT integration methods and traffic volumes are suggested. Three different integration methods (*VPW*, *RWY*, *VPE*) are demonstrated and performed within an FTS for Hamburg airport using the software AirTOP. The results describe the basis for evaluating the integration method based on delay and energy requirements, with respect to the impact on conventional air traffic.

### 4.1. Punctuality Results

Evaluating all simulation runs, the greatest impact was observed on the conventional air traffic side due to the *RWY* integration. The ATs occupy the conventional runway system temporarily by merging into the final approach and further usage of the runway itself. The flight performance of an AT and a conventional aircraft shows differences of more than 100 kt regarding approach speed, which creates a heterogeneous mix in the final approach. To avoid violation of separation minima and runway incursions, greater separations are applied causing the observed delays. With an increased traffic volume, this effect amplifies, especially during the peak hours. The distribution of delays among the aircraft shows that

arrivals are especially affected by the *RWY* integration, which strengthens the described effect due to delay. Although a late integration point was designed to keep this effect as low as possible, eight delayed aircraft occured with five ATs integrated per hour. Departure operations play a minor role within the total delay for the *RWY* integration method. A further analysis showed that the most of the delay arose during the final approach.

The integration at *VPW* creates fewer delayed aircraft because of lower dependencies on the conventional air traffic. Departures were affected due to the designed AT flight path and the dependency on the jet blast area. However, the impact was less, compared to that of the *RWY* integration. Hence, only a few aircraft show a delay equal to or greater than 15 min.

A lower number of delayed ATs within the *VPWAT20* scenario compared to the *VPEAT20* scenario was found in the simulation methodology. The reason is that the ATs were partially integrated in blocks to take advantage of the lower separation minima between ATs, so that less overall delay occurred. This effect was not present in earlier scenarios.

The integration through *VPE* created, as claimed, almost no delay for the aircraft. The delays that appear were equally distributed between arrivals and departures.

### 4.2. Energy Results

The results of the energy analysis from Section 3.2 pointed out that the *RWY* integration leads to the highest energy consumption among the ATs. This effect is caused by various dependencies between the aircraft and ATs, leading to holding procedures for the ATs. As a result, the flight time increases and the ATs use more electric energy. Within the *VPW* integration, the ATs have the lowest overall energy consumption. This can be explained by the layout of the route structure. Most of the city TLOF-vertiports are located in the south of the airport. Thus, the connection to the *VPW* is the shortest possible integration method for those operations.

Regarding the battery capacity analysis, only the *RWY* integration causes AT operations to exceed the available battery capacity. The only anomaly was found for the *RWYAT20* scenario, which has fewer affected ATs than the *RWYAT15* scenario. This deviation occurred primarily during a specific period of time because of the modelling characteristics regarding the FCFS specified in Section 2.2. As a consequence of the FCFS, one of the two traffic flows had to go into a holding pattern to achieve sufficient separation. Regarding the *RWYAT15* scenario, the conventional air traffic arrived first at the specific point on the approach path during the specific period of time, which resulted in a holding pattern for the ATs. These ATs were not affected within the *RWYAT20* scenario during the identical period of time, since the increased number of ATs led to the situation in which ATs already had to go into the holding pattern to create sufficient separation among each other. Therefore, the FTS decided that the conventional air traffic also had to go into a holding to respect the FCFS (AT in holding arrived first), and this led to the situation in which the ATs received their landing clearance prior to the conventional air traffic. In summary, a maximum number of 19 ATs (*RWYAT15*) used more energy than available. This is a serious safety issue, which confirms that the battery capacity constitutes a bottleneck for the *RWY* integration.

Usually, aircraft are carrying additional fuel for contingency and other safety reasons, which should also apply for ATs. Regarding ICAO Annex 6 [37], aircraft have to carry additional fuel to fly for 30 or 45 min (depending on the engine type) in a holding pattern above the aerodrome. In terms of the ATs, the range of 50 km (cf. Table 3) is much shorter compared to conventional aircraft. Therefore, an energy reserve up to 45 min would be unrealistic for this type of AT. As a first estimation, an energy reserve of 10% of the battery capacity is assumed to be necessary. Figure 10 shows the results of ATs that exceed this energy reserve. It is apparent that the *RWY* integration still has the most affected ATs with a maximum number of 26 ATs (*RWYAT20*). Additionally, the *VPW* integration has some ATs that cannot meet the defined energy reserve in any of the investigated traffic volumes.

The primary reason is the extension of departures northwards (cf. Section 2.1). The route distance from *VPW* to the northern vertiport is slightly longer compared to *VPE*, which increases the energy consumption.

In summary, the energy analysis shows that the *VPE* integration is the preferred solution regarding the impact of ATs on the aircraft fuel consumption. The *VPW* integration leads to the lowest overall energy consumption for the ATs, but it requires a high energy consumption for departures heading towards vertiports located north. Since the ATs' power characteristics are simplified assumptions, this analysis may have some limitations. However, the results allow an assessment of the three integration methods.

*4.3. Summary*

Overall, the developed model and results show that the integration is possible within the defined punctuality criteria and energy requirements. Only the integration into the final approach for the *RWY* method violates the punctuality criteria. Yet an integration of five ATs per hour with a contemporaneous maximum amount of 41 aircraft movements (using a conventional flight plan) exceeds the criteria. The integration of ATs at airports with low or even no dependencies on the conventional air traffic offers new potentials for airports.

The integration via *VPE* provides the lowest average delay and energy consumption over all simulation runs. Additionally, the scalability potential of the chosen parking level can be addressed. In comparison, this possibility is not given, assuming a (*VPW*) due to limited space. Likewise, the *VPE* enables the highest flexibility over all integration methods for the flight path design. When considering the requirement of a minimum angle of 15° to conventional air traffic, considerable airspace can be used for flexible routing. This could be of advantage in noise reduction through distribution possibilities. Nevertheless, there is potential for optimisation in route definition regarding an exemplary final approach towards the vertiport. However, the fact that an implementation of a TLOF-vertiport requires additional costs due to infrastructural expansion or further costs for staff, such as turn around (TA) and security personnel or transfer facilities, should not be disregarded. The integration via *RWY* might provide a reasonable and cost-effective method for low-traffic hours, because it does not require additional infrastructure or staff.

The analysed model and the data shown provide specific results for Hamburg airport, which provides unique circumstances such as the intersecting two runway configuration, the runway mode and the location within the city (north-east of the city centre). However, the runway configuration has an influence on the conducted results, which was not part of this study. Further studies will quantify their impact on the capacity and energy results.

**5. Conclusions**

The present study designs a concept and analysis approach for integrating ATs at an airport. It suggests steps to implement AT operations at a desired airport and analyses and provides a quantification of the capacity impact and energy consumption. Integration into conventional runway systems is only recommended for low-traffic hours or at small single runway airports, since the results showed high effects of ATs on the conventional traffic delay for a high traffic airport. During rush hours, the integration should be carried out on distinct TLOF-vertiports apart from the runway. Hence, airports with medium- or high-traffic volume are advised to have at least two possible TLOF opportunities for ATs, which can be switched regarding the current traffic load and runway mode.

The analysed example of Hamburg airport has shown that the intersecting runway system especially causes difficulties for implementing AT operations at that airport. Crucial points might be the the necessary crossing or integration of the final approach for most airport systems, due to the desired destination points, close operations near or in the direction of the runway extensions and merging points due to separating AT traffic from conventional air traffic. The results have shown that, already, five ATs' integration into the final approach causes unjustifiable delays. Therefore, TLOF-vertiports are advisable with

a large angle between the city centre and the approach and departure flight path. With a minimum angle of 15°, simultaneous operations can be performed from ATs and aircraft. This could improve cost-effectiveness and overall global operations at an airport including AT operations. If such a location is not feasible, it is recommended that, for reasons of security, a fly around or low level flying under the final approach might provide the most reliable and convenient solution for crossing. ATs and their flight performances are not yet certified and introduce uncertainties regarding their behaviour to cross a final approach only with vertical separation. The handling of these uncertainties and the potential of shortcuts, depending on the operational situation, should be evaluated in human-in-the-loop simulations with air traffic controllers. Thereby, it can be assessed whether the additional effort of AT integration is feasible for air traffic control as well as whether controllers are able to make final approach crossings more efficient.

The impact of ground operations (TA and luggage process) causing delays should not be neglected and has not yet been analysed. For time considerations and the cost effectiveness of vertiports, separate security checks, especially on landside vertiports, could represent an important factor Therefore, a detailed ground model, covering TA and security checks, will expand the developed model in an ongoing study, quantifying their impact on the results and timing.

**Author Contributions:** Conceptualization, N.A. and O.P.; methodology, N.A. and O.P.; software, O.P.; validation, N.A., O.P. and S.S.-M.; formal analysis, N.A.; investigation, N.A.; writing—original draft preparation, N.A. and O.P.; writing—review and editing, S.S.-M.; visualization, N.A. and O.P.; supervision, S.S.-M. All authors have read and agreed to the published version of the manuscript.

**Funding:** This research received no external funding.

**Institutional Review Board Statement:** Not applicable.

**Informed Consent Statement:** Not applicable.

**Data Availability Statement:** Data can be available by contacting the corresponding author, Nils Ahrenhold.

**Conflicts of Interest:** The authors declare no conflict of interest.

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
