# Peer review of "Impact of Air Taxis on Air Traffic in the Vicinity of Airports"

_infrastructures, doi:10.3390/infrastructures6100140_

Round 1

Reviewer 1 Report

The subject discussed in the article is important for addressing the urban traffic density. The paper correctly defines and addresses the problem, providing an interesting methodology to study the implementation of air taxis as an urban transport mode. Results and analysis are well presented and clearly organized. I would recommend the publication of the study. Just two minor comments:

  1. The authors could explain why the hypothesis of neglecting the wind (Line 248) is realistic in their study
  2. From an environmental point of view, the authors considered only the energy consumption. Do not they think that the noise is also an important issue? It is especially relevant because they are dealing with urban environment.

Author Response

Dear Reviewer,

Thank you for considering our manuscript Impact of Air Taxis on Air Traffic in the Vicinity of Airports for publication in the special issue "Air Traffic Management: Airport Operations" of the MDPI Infrastructures journal.

We appreciate the reviewers´ effort in reviewing our manuscript and their constructive comments. According to their suggestions, we revised the manuscript carefully. In the following we will provide a point-by-point response to the reviewer´s comments.

  1. The authors could explain why the hypothesis of neglecting the wind (Line 248) is realistic in their study

    This study demonstrates a first estimation under best weather conditions. Currently there are no valid data for crosswind behavior and susceptibility for our multicopter. We extended the description of this hypothesis in our manuscript.
  2. From an environmental point of view, the authors considered only the energy consumption. Do not they think that the noise is also an important issue? It is especially relevant because they are dealing with urban environment.

    In this study we focus on the treatment of fuel and energy consumption within the simulation as one of the environmental factors. Noise and other environmental factors will be part of a further study. A more detailed explanation is included in our manuscript.

We hope to have revised our manuscript to your satisfaction and look forward to your response considering the further processing.

Sincerely,

Nils Ahrenhold

Reviewer 2 Report

The submission addresses an interesting problem and is well written, in general. But, I have several concerns with this submission, which should be addressed before recommending publication:

- The authors are solving a quite standard problem, what is called a location/routing problem in the literature. And there exist also standard solution techniques, including exact techniques for smaller instances. The authors solve the problem using a simulation approach, and it is not clear up to what degree these experiments lead to close-to-optimal solutions.

- A fixed runway configuration is used. The results should depend ont he actual configuration - which changes even multiple times a day. How sensitive are the results towards these changes?

- How is the security check conducted and how luggage processed? Although it is written in the Conclusions that ground processing is not considered ... the additional overhead of these two steps should considerable, and has an effect on the results/timings.

- AT is expected to be largely on-demand driven... how can one reliably simulate AT operations? The authors should better discuss the origin of demand data and how it is subject to such on-demand concerns.

- The main finding is that if you place the integration point in the wrong location (RWY), then there will be a lot of conflicts, more delay, and more emissions. This is kind of expected. The authors should better highlight the more generic findings. This is done up to some degrees (e.g., regarding the crossing point of runways), but not fully convincing yet.

- I would rename "integration methods" to "integration locations" or "integration points"; as the choice of location is not really a method.

- I suggest to expand the reference section and discussion of air taxi related literature. Most reference right now are technical reports and other documents. There has been tremendous work in this area recently, which should be put into context. For instance, the authors can check the following studies and references therein.

S Rath et al. Air taxi skyport location problem for airport access. arXiv preprint arXiv:1904.01497, 2019

X Sun et al. Operational Considerations Regarding On-Demand Air Mobility: Literature Review and Research Challenges. Journal of Advanced Transportation pp. 3591034, 2021

- There are a few easy-to-fix language issues: prevent a feasible fly->flight, section X->Section X

Author Response

Dear Reviewer,

Thank you for considering our manuscript Impact of Air Taxis on Air Traffic in the Vicinity of Airports for publication in the special issue "Air Traffic Management: Airport Operations" of the MDPI Infrastructures journal.

We appreciate the reviewers´ effort in reviewing our manuscript and their constructive comments. According to their suggestions, we revised the manuscript carefully. In the following we will provide a point-by-point response to the reviewer´s comments.

  1. The authors are solving a quite standard problem, what is called a location/routing problem in the literature. And there exist also standard solution techniques, including exact techniques for smaller instances. The authors solve the problem using a simulation approach, and it is not clear up to what degree these experiments lead to close-to-optimal solutions.

    This study does not focus routing problem, but route definition is part of the conducted impact analysis. The study developed a system to measure the impact of air taxis on airside capacity of an airport. The optimization potential of the route definition will be qualified by a further study.

  2.  A fixed runway configuration is used. The results should depend on the actual configuration - which changes even multiple times a day. How sensitive are the results towards these changes?

    The results surely depend on the runway configuration. For this study the most used runway configuration in Hamburg is used and explained more detailed in the revised manuscript (line 262). A sensitivity study is part of a connected study. In the discussion part we point this out (line 556).

  3. How is the security check conducted and how luggage processed? Although it is written in the Conclusions that ground processing is not considered ... the additional overhead of these two steps should considerable, and has an effect on the results/timings.

    Within the simulation environment (AirTOP), the security check and luggage processes are not included. Quantification of this processes take place as part of a subsequent thesis. (line 589)

  4. AT is expected to be largely on-demand driven... how can one reliably simulate AT operations? The authors should better discuss the origin of demand data and how it is subject to such on-demand concerns.

    A more detailed explanation is included in the revised manuscript. The on-demand AT service is directly linked to the conventional flight plan. AT are requested regarding the departure and arrival times of conventional aircraft.

  5. The main finding is that if you place the integration point in the wrong location (RWY), then there will be a lot of conflicts, more delay, and more emissions. This is kind of expected. The authors should better highlight the more generic findings. This is done up to some degrees (e.g., regarding the crossing point of runways), but not fully convincing yet.

    Expectations confirmed, but a quantification was necessary. Results show that already five air taxis generate problems for the runway integration.

  6.  I would rename "integration methods" to "integration locations" or "integration points"; as the choice of location is not really a method.

    A definition for integration method is included in the revised manuscript. The term method is used, since a location and procedures are included (line 151).

  7. I suggest to expand the reference section and discussion of air taxi related literature. Most reference right now are technical reports and other documents. There has been tremendous work in this area recently, which should be put into context. For instance, the authors can check the following studies and references therein.

    The reference section and discussion of air taxi related literature was expanded.

  8. There are a few easy-to-fix language issues: prevent a feasible fly->flight, section X->Section X

    Issues were corrected.

We hope to have revised our manuscript to your satisfaction and look forward to your response considering the further processing.

Sincerely,

Nils Ahrenhold

Reviewer 3 Report

The manuscript "IMPACT OF AIR TAXIS ON AIR TRAFFIC IN THE VICINITY OF AIRPORTS" is interesting and within the scope of publication of this journal, I suggest some minor corrections:

a) See the format of this journal, missing the top image and other information, this is really important;
b) The abstract must present more clearly some numerical data and highlights of your research;
c) Scientific innovation and contribution must be highlighted after the objectives, at the end of the introduction;
d) There are few references, you should stick and look for more scientific articles in the research area to complement the state of the art of your research;
e) Observe and revise some figures, text quality is low in some cases.

Author Response

Dear Reviewer,

Thank you for considering our manuscript Impact of Air Taxis on Air Traffic in the Vicinity of Airports for publication in the special issue "Air Traffic Management: Airport Operations" of the MDPI Infrastructures journal.

We appreciate the reviewers´ effort in reviewing our manuscript and their constructive comments. According to their suggestions, we revised the manuscript carefully. In the following we will provide a point-by-point response to the reviewer´s comments.

  1. See the format of this journal, missing the top image and other information, this is really important;

    We used the official MDPI Overleaf template. The top image appears after acceptance, to our current knowledge.

  2. The abstract must present more clearly some numerical data and highlights of your research;

    The abstract was revised and numerical data included.

  3. Scientific innovation and contribution must be highlighted after the objectives, at the end of the introduction;

    Aim and focus of the study is included after objectives. (line 123)

  4. There are few references, you should stick and look for more scientific articles in the research area to complement the state of the art of your research;

    The reference section was expanded with more air taxi related literature.
  5. Observe and revise some figures, text quality is low in some cases.

    Figures for the punctuality results were included separately to increase text quality.

We hope to have revised our manuscript to your satisfaction and look forward to your response considering the further processing.

Sincerely,

Nils Ahrenhold